# Simultaneous Wireless Information and Power Transfer in Multi-User OFDMA Networks with Physical Secrecy

**DOI:** 10.3390/s22103814

**Published:** 2022-05-18

**Authors:** Pubet Sangmahamad, Kampol Woradit, Poompat Saengudomlert

**Affiliations:** 1Department of Electronics and Telecommunication Engineering, Faculty of Engineering, Rajamangala University of Technology Thanyaburi, Pathum Thani 12110, Thailand; pubet.s@en.rmutt.ac.th; 2Optimization Theory and Applications for Engineering Systems Research Group, Department of Computer Engineering, Faculty of Engineering, Chiang Mai University, Chiang Mai 50200, Thailand; 3Bangkok University Center of Research in Optoelectronics, Communications and Computational Systems, School of Engineering, Bangkok University, Pathum Thani 12120, Thailand; poompat.s@bu.ac.th

**Keywords:** secrecy rate, subcarrier allocation, power splitting ratio, energy harvest, wireless powered communications, simultaneous wireless information and power transfer

## Abstract

This paper considers simultaneous wireless information and power transfer (SWIPT) from a base station to multiple Internet of Things (IoT) nodes via orthogonal frequency-division multiple access (OFDMA), where every node can eavesdrop on the subcarriers allocated to other nodes. Application layer encryption is unsuitable for IoT nodes relying on energy harvesting, and physical layer secrecy should be deployed. The different channels among users on every subcarrier can be exploited to obtain physical layer secrecy without using artificial noise. We propose an algorithm to maximize the secrecy rate of IoT nodes by jointly optimizing the power splitting ratio and subcarrier allocation. For fairness, the lowest total secrecy rate among users is maximized. Through simulations, the proposed algorithm is compared with the minimum effort approach, which allocates each subcarrier to the strongest node and selects the minimum sufficient power splitting ratio. The obtained secrecy rate is 3 times (4.5 over 1.5 bps/Hz) higher than that of the minimum effort approach in every case of parameters: the base station’s transmit power, the minimum harvested energy requirement of an IoT node and the energy harvesting efficiency.

## 1. Introduction

Densely deploying many Internet of Things (IoT) nodes is challenging for medium access control and power supply, where throughput drops with more users and battery replacement incurs burden and cost. The former can be mitigated by efficient protocols [1,2,3]. A solution for the latter is energy harvesting [4], especially simultaneous wireless information and power transfer (SWIPT) [5], where receivers harvest energy and decode information simultaneously from the same received signal by a power splitter. This power splitter divides the received signal into two copies. Their power ratio between two copies is referred to as the power splitting ratio.

SWIPT was first studied in point-to-point communication, and both transmitter and receiver have a single antenna. Later, SWIPT with cooperative communications is optimized in terms of throughput using convex optimization problem formulation in [6], where the network consists of single-antenna nodes: source, relay and destination. The solution is simple but does not support multi-user cases. SWIPT with multiple-input multiple-output (MIMO) is considered in [7], where the beamforming and power splitting ratio are adapted based on limited feedback from the receivers to maximize rate and harvested energy. Even though multi-user is considered, MIMO incurs high complexity. In practice, IoT devices have a single antenna and do not support high complexity computation. SWIPT is combined with time switching in [8], where the receiver can switch to either information decoding mode or energy harvesting mode within a transmission period. However, the time switching technique has difficulty in implementation.

Orthogonal frequency-division multiple access (OFDMA) is a multi-carrier transmission technique where all subcarriers are orthogonal and allocated to multiple users, where any subcarrier subset can be allocated to a user. The combination of the OFDMA network with the SWIPT system was proposed in [9], where all nodes have a single antenna. The distributed antenna system [10] is incorporated in OFDM [11] to improve the energy efficiency. In [12], the energy efficiency of OFDM is maximized in a downlink point-to-point SWIPT system by optimizing power allocation.

Practical IoT devices do not support complex encryption for information security, while physical secrecy is promising because the complex part is with the base station [13]. Basically, the signal-to-noise ratio (SNR) at a legitimate receiver must be higher than that at eavesdroppers to obtain the rate, called secrecy rate, that only the legitimate receiver can decode the information [14]. Secrecy rate is the difference between the legitimate receiver’s rate and the eavesdropper’s rate [15]. In the case of multiple eavesdroppers, the secrecy rate is the difference between the rate of the legitimate receiver and the rate of the eavesdropper with the highest SNR. In [16], the SNR of an eavesdropper is deducted by deploying a jammer node to send an artificial noise, but the artificial noise also affects the SNR of the legitimate receiver.

Artificial noise does not only interfere with eavesdroppers but also serves as the energy source. Thus, artificial noise is combined with SWIPT [17], where the transmit power allocation and power splitting ratio are jointly optimized to maximize the secrecy rate in the SWIPT system with a single antenna. However, the considered eavesdropper is the energy harvesting unit within the receiver, not another receiver. An eavesdropper, which is a separated receiver, is considered in [18], where the transmitter has multiple antennas to steer the beam of artificial noise to the eavesdropper, and the transmit power is optimally split into two portions: artificial noise and information. This idea is extended to the case of multiple eavesdroppers in [19], where the power allocation must be optimized over multiple beams of artificial noise to the eavesdroppers.

The artificial noise for SWIPT is combined with OFDMA in [20,21], where transmit power of each subcarrier and subcarrier allocation are optimized to maximize the weighted sum secrecy rate of multiple users. With multi-user OFDMA, the optimization problem becomes a mixed-integer programming problem, which is non-convex. The dual problem is considered instead based on the Lagrange duality method and a proposed suboptimal algorithm.

The approach, which does not rely on artificial noise, is introduced in [22]. It considers multi-user SWIPT OFDMA, where all receivers and a transmitter have a single antenna and are eavesdropped by an eavesdropper. The sum harvesting power of all users is maximized by jointly optimizing subcarrier allocation and power splitting ratio selection under a constraint that every user’s secrecy rate meets the minimum requirement. The problem is NP-hard. As a result, the subcarrier allocation and power splitting ratio selection are alternately optimized by an iterative algorithm, which fixes one of them while finding the best of another, and the suboptimal solution is obtained.

From the literature above, it is assumed that the eavesdropper is a node outside the network. In fact, an eavesdropper can be an insider [23]. Users in the same network know the protocol and can eavesdrop on each other. Moreover, in the case of IoT devices with SWIPT, they locate in the vicinity of the transmitter and experience the same path loss and shadowing. Only fast fading causes the rate difference, which gives a non-zero secrecy rate. When OFDMA is used, any user can eavesdrop on the subcarriers of other users. This paper investigates this scenario to find the maximum secrecy rate that can be obtained.

Specifically, we consider a multi-user SWIPT OFDMA network, which comprises a multi-antenna transmitter and multiple single-antenna receivers (users). Every user can eavesdrop on the subcarriers of other users. Still, the transmitter can control the subcarrier allocation and the power splitting ratio of every user to ensure that every user has a non-zero total secrecy rate (sum secrecy rate of all allocated subcarriers). Without artificial noise, the secrecy rate is maximized by jointly optimizing the subcarrier allocation and power splitting ratios of all users under the constraint that the harvested energy of every user must meet the minimum requirement. For fairness, the secrecy rate in this place is the lowest total secrecy rate among users. Since the formulated problem is a non-linear mixed-integer programming problem, quantization is introduced to find the optimal solution. The results indicate that there exists a secrecy rate ceil, which depends on three factors: transmit power, minimum required harvesting energy and energy harvesting efficiency.

This paper is organized as follows. Section 2 describes the system and network models. Section 3 proposes the algorithm to find the optimal solution. Section 4 presents the simulation results and discussions. Section 5 gives the conclusion.

## 2. System and Network Models

This section describes the system and network models, where we consider a downlink multiple-input single-output (MISO) multi-user OFDMA network based on SWIPT as illustrated in Figure 1. The network consists of a multi-antenna transmitter, *K* single-antenna receivers (users), and *N* subcarriers. The transmitter is equipped with M>1 antennas and N≥K. The transmitter has a transmit power of $P_{t}$, which is allocated equally to every subcarrier; that is, every subcarrier has a transmit power of $P_{t}/N$. Power allocation is not optimized in this paper because the problem is non-convex and needs simplification to be tractable [24] while we aim to provide results for benchmark. When the transmitter transmits an OFDM signal to all users, each user can harvest energy and decode information from the received signal with a power splitter based on the SWIPT receiver model as shown in Figure 2. A portion of ρ is harvested for energy, while the left 1−ρ is decoded for information. In the information portion, each user can eavesdrop on the subcarriers of other users.

The bandwidth of OFDMA is equally divided into *N* channels, each of which associates with a subcarrier. The set of subcarriers is denoted by N (N={0,1,…,N−1}). Each subcarrier must be allocated to only one user. To keep the problem tractable, we assume that the IoT node mobility is relatively slow compared with a frame rate of OFDMA. Hence, the channel is quasi-static Rayleigh frequency-selective fading. Every subcarrier’s channel state information (CSI) is constant for a transmission period and independently varies in the next period. The CSIs of all subcarriers and users are assumed to be perfectly known at the transmitter. Let *m*, *k* and *n* index the transmit antenna, the user and the subcarrier, respectively. The channels of all users are independent and identically distributed (i.i.d.). The discrete-time channel impulse response from the *m*th transmit antenna to the *k*th user can be expressed as
(1)$$h_{m,k}\left(\tau \right)=\sum_{l=0}^{L-1}\alpha_{m,k}\left(l\right)\delta \left(\tau -l\right),$$
where *L* is defined as the number of channel delay taps and the *l*th tap is modeled as $\alpha_{m,k}\left(l\right)∼\mathrm{CN}\left(0,\delta_{l}^{2}\right)$. The power delay profile is normalized by
(2)$$\sum_{l=0}^{L-1}\delta_{l}^{2}=1.$$

The channel frequency response at the *n*th subcarrier (n∈N) between the *m*th transmit antenna and the *k*th user can be expressed as
(3)$$H_{m,k}\left(n\right)=\sum_{l=0}^{L-1}\alpha_{m,k}\left(l\right)e^{-j2\pi nl/N},n\in N$$

At the *k*th user, the received signal is split into two portions: the first portion for energy harvesting and another portion for information decoding. The ratio between two portions is determined by $\rho_{k}$, where $\rho_{k}\in \left[0,1\right]$. Hence, a $\rho_{k}$ portion of the received signal is used for energy harvesting, and the remaining $1-\rho_{k}$ portion is used for information decoding as illustrated in Figure 2.

In the first portion, the conversion efficiency of the energy harvesting unit at every user is not 100% and is denoted by ξ, where 0≤ξ≤1. At every user, the energy is harvested from all subcarriers. The transmitter exploits the full CSI of all users to do *M*-antenna beamforming on each subcarrier to maximize the instantaneous SNR of the user, to which that subcarrier is allocated. Given that the *n*th subcarrier is allocated to the $k^{\prime }\left(n\right)$th user, the total harvested energy of the *k*th user is obtained by
(4)$$E_{k}=\xi \rho_{k}\frac{P_{t}}{N}\sum_{n=0}^{N-1}{\left|\sum_{m=1}^{M}H_{m,k^{\prime }}\left(n\right)\frac{H_{m,k^{\prime }\left(n\right)}^{*}\left(n\right)}{\sqrt{\sum_{m^{\prime }=1}^{M}{\left|H_{m,k^{\prime }\left(n\right)}^{*}\left(n\right)\right|}^{2}}}\right|}^{2}+\xi \sigma^{2},$$
where $\sigma^{2}$ is the variance of additive white Gaussian noise, which combines the antenna noise $\sigma_{a}^{2}$ and the signal processing noise $\sigma_{z}^{2}$, that is, $\sigma^{2}=\sigma_{a}^{2}+\sigma_{z}^{2}$. Note that the conversion efficiency ξ lumps a factor that converts the unit of power to the unit of energy.

In the second portion, each user decodes the information conveyed by its allocated subcarriers. In case that no user tries to eavesdrop on other subcarriers, the transmitter exploits the full CSI of all users to do *M*-antenna beamforming on each subcarrier to maximize the instantaneous information rate of the user, allocated on that subcarrier. Suppose that the *n* subcarrier is allocated to the *k*th user, the instantaneous information rate of the *k*th user at the *n* subcarrier is given by
(5)$$\begin{array}{c} \begin{array}{cc} r_{k}\left(n\right) & =\log_{2}\left(1+\frac{P_{t}\left(1-\rho_{k}\right)}{NM\sigma^{2}}{\left|\sum_{m=1}^{M}\frac{H_{m,k}\left(n\right)H_{m,k}^{*}\left(n\right)}{\sqrt{\sum_{m=1}^{M}{\left|H_{m,k}^{*}\left(n\right)\right|}^{2}}}\right|}^{2}\right)\\ \; & =\log_{2}\left(1+\frac{P_{t}\left(1-\rho_{k}\right)}{N\sigma^{2}}{\left|\frac{\sum_{m=1}^{M}{\left|H_{m,k}\left(n\right)\right|}^{2}}{\sqrt{\sum_{m=1}^{M}{\left|H_{m,k}^{*}\left(n\right)\right|}^{2}}}\right|}^{2}\right)\\ \; & =\log_{2}\left(1+\frac{P_{t}\left(1-\rho_{k}\right)}{N\sigma^{2}}\frac{{\left(\sum_{m=1}^{M}{\left|H_{m,k}\left(n\right)\right|}^{2}\right)}^{2}}{\sum_{m=1}^{M}{\left|H_{m,k}^{*}\left(n\right)\right|}^{2}}\right)\\ \; & =\log_{2}\left(1+\frac{P_{t}\left(1-\rho_{k}\right)}{NM\sigma^{2}}\sum_{m=1}^{M}{\left|H_{m,k}\left(n\right)\right|}^{2}\right). \end{array} \end{array}$$

In this paper, every user can eavesdrop on other subcarriers. The instantaneous eavesdropping information rate of the $k^{\prime }$th user ($k\neq k^{\prime }$) at the *n* subcarrier is given by
(6)$$\begin{array}{c} \begin{array}{cc} r_{k^{\prime }}\left(n\right) & =\log_{2}\left(1+\frac{P_{t}\left(1-\rho_{k^{\prime }}\right)}{N\sigma^{2}}{\left|\sum_{m=1}^{M}\frac{H_{m,k^{\prime }}\left(n\right)H_{m,k}^{*}\left(n\right)}{\sqrt{\sum_{m=1}^{M}{\left|H_{m,k}^{*}\left(n\right)\right|}^{2}}}\right|}^{2}\right)\\ \; & =\log_{2}\left(1+\frac{P_{t}\left(1-\rho_{k^{\prime }}\right)}{N\sigma^{2}}\frac{{\left|\sum_{m=1}^{M}H_{m,k^{\prime }}\left(n\right)H_{m,k}^{*}\left(n\right)\right|}^{2}}{\sum_{m=1}^{M}{\left|H_{m,k}^{*}\left(n\right)\right|}^{2}}\right). \end{array} \end{array}$$

The achievable secrecy rate of the *k*th user at the *n*th subcarrier is defined as the non-negative difference between the rate of the *k*th user and the maximum rate of all other users and can be written as
(7)$$R_{k,n}^{s}={\left(r_{k}\left(n\right)-\max_{k^{\prime }\neq k}r_{k^{\prime }}\left(n\right)\right)}^{+},$$
where ${\left(\cdot \right)}^{+}≜\max\left(\cdot ,0\right)$. More than one subcarriers can be allocated to a user. The total secrecy rate of a user is the summation of secrecy rate of all subcarriers allocated to that user. The total secrecy rate of the user with the lowest total secrecy rate can be calculated by
(8)$$R^{s}=\min_{k}\sum_{n=0}^{N-1}x_{k,n}R_{k,n}^{s},$$
where $x_{k,n}$ denotes the indicator function of the subcarrier allocation and is defined by
(9)$$x_{k,n}=\left\{\begin{array}{c} 1,\;n^{th}\mathrm{subcarrier}\;\mathrm{is}\;\mathrm{allocated}\;\mathrm{to}\;\mathrm{the}\;k^{th}\mathrm{user},\\ 0,\;\mathrm{otherwise}. \end{array}\right.$$

## 3. Proposed Algorithm

This section presents the algorithm to find the optimal subcarrier allocation and power splitting ratios for maximizing the total secrecy rate of the user with the lowest total secrecy rate. The optimization problem can be formulated as
(10)$$\begin{array}{ccc} \underset{x_{1,0},\ldots ,x_{K,N-1},\rho_{1},\ldots ,\rho_{K}}{\mathrm{maximize}} & R^{s} & \end{array}$$
(11)$$\begin{array}{ccc} \mathrm{subject}\;\mathrm{to} & \sum_{k=1}^{K}x_{k,n}\leq 1, & \;\forall n \end{array}$$
(12)$$\begin{array}{ccc} \; & x_{k,n}\in \left\{0,1\right\}, & \;\forall k,n \end{array}$$
(13)$$\begin{array}{ccc} \; & 0\leq \rho_{k}\leq 1, & \;\forall k \end{array}$$
(14)$$\begin{array}{ccc} \; & E_{k}\geq \bar{E}, & \;\forall k, \end{array}$$
where $x_{1,0},\ldots ,x_{K,N-1}$ are the indicator functions of subcarrier allocation for all users and subcarriers, and $\rho_{1},\ldots ,\rho_{K}$ are the power splitting ratios for all users. Each subcarrier can only be used by one user as determined by constraints (Equation 11) and (Equation 12). The lower bound and upper bound every power splitting ratio is defined by constraint (Equation 13). The harvested energy at each user must be greater than or equal to the minimum required harvesting energy $\left(E_{k}\geq \bar{E}\right)$ to operate the receiver as determined by constraint (Equation 14), which is assumed to be the same for all users. The algorithm for solving the formulated optimization problem is given in Algorithm 1.

The algorithm is calculated for each realization of a channel impulse response. Optimizing the indicator functions and the power splitting ratios is separated into outer loops and inner loops. The outer loops try all $K^{N}$ possible $x_{1,0},\ldots ,x_{K,N-1}$, which are binary. For each outer loop, $\rho_{1},\ldots ,\rho_{K}$, which are real numbers between 0 and 1, are optimized. Since the problem is non-convex, a closed-form solution cannot be found. The optimal $\rho_{1},\ldots ,\rho_{K}$ are found numerically by quantizing the interval [0,1] for every $\rho_{k}$. Then, all quantized $\rho_{1},\ldots ,\rho_{K}$ are tried as the inner loops. The step size of quantization determines the number of inner loops. The small quantization step size gives an accurate solution, but the number of inner loops becomes large. The number of inner loops also becomes large when the number of users or subcarriers is large. In that case, the Monte Carlo method can be applied instead of quantization.

To take into account the constraint (Equation 14), if every user obtains the harvested energy, greater than or equal to $\bar{E}$ in an inner loop, then $R^{s}$ will be calculated and stored for later comparisons. Otherwise, the harvested energy is not sufficient for every user, and the $R^{s}$ is not calculated. After all outer loops and inner loops are computed, all stored $R^{s}$ are compared to find the maximum $R^{s}$. The optimizer that associates with the maximum $R^{s}$ is the optimal solution, denoted by $x_{1,0}^{*},\ldots ,x_{K,N-1}^{*},\rho_{1}^{*},\ldots ,\rho_{K}^{*}$. If the set of stored $R^{s}$ is a null set, a communication outage occurs with that channel impulse response realization, and the obtained secrecy rate per user becomes zero. The average total secrecy rate of the user with the lowest total secrecy rate is presented in the next section.
**Algorithm 1** Maximizing the lowest total secrecy rate in a realization1:randomize $h_{m,k}\left(\tau \right)$ for all m,k2:compute $H_{m,k}\left(n\right)$ for all m,k with (Equation 3)3:**for** each trial of $x_{1,0},\ldots ,x_{K,N-1}$ **do**4:   given $x_{1,0},\ldots ,x_{K,N-1}$, compute beamforming for every subcarrier5:   **for** each trial of $\rho_{1},\ldots ,\rho_{K}$ **do**6:     given $\rho_{1},\ldots ,\rho_{K}$, compute $E_{1},\ldots ,E_{K}$ with (Equation 4)7:     **if** $E_{k}\geq \bar{E},\forall k$ **then**8:        compute $r_{k}\left(n\right),\forall k,n$ with (Equation 5) and (Equation 6)9:        compute and store secrecy rate $R^{s}$ with (Equation 7) and (Equation 8)10:     **else**11:        skip this trial due to energy outage12:     **end if**13:   **end for**14:**end for**15:**if** the set of stored $R^{s}$ is not null **then**16:   compare all stored $R^{s}$ to find the maximum value17:   $x_{1,0},\ldots ,x_{K,N-1}$ and $\rho_{1},\ldots ,\rho_{K}$ that gives the maximum $R^{s}$ is the solution18:**else**19:   the communication outage occurs in this realization20:**end if**

## 4. Simulation Results and Discussions

This section presents the computer simulation results of the proposed algorithms. The simulations were conducted with the parameters in Table 1. The power delay profile of the channel impulse responses is uniform. A transmit power of 1 W (30 dBm) is typically available for access points, and a receiver can receive an information with a power sensitivity of −60 dBm [25], which was chosen to be the noise variance in this paper. Self-resonant coils, which are commonly implemented for wireless power transfer, provides an efficiency of 40% in experimentation when a distance between the transmitter and receiver is about 2 m [26]. Therefore, the energy harvesting efficiency is set at 0.4. The real test in [27] shows that a smart IoT node processes a pattern recognition with a power of 11.84 mW. Hence, the minimum required harvesting energy here is set at 10 mW.

In realistic environment, subcarriers have unequal noise variances, which affect the total secrecy rate of users. This paper aims to provide an insight of the parameters that can be adjusted by the system, namely, transmit power, energy harvesting efficiency and minimum required harvesting energy. Equal noise variance is assumed to exclude the effect of noise variances, which depends on environment. Still, simulations with unequal noise variances can be conducted by modifying the total harvested energy in (Equation 4) and the instantaneous information rate in (Equation 5) and (Equation 6). Other equations and the proposed algorithm do not need to be changed.

We first compare the proposed algorithm, which is optimum, with the minimum effort approach, which is non-optimum. The minimum effort approach allocates each subcarrier to the user, who has the highest instantaneous SNR in that subcarrier and set the power splitting ratio of each user to meet the minimum required harvesting energy, that is,
(15)$$x_{k,n}^{*}=\left\{\begin{array}{c} 1,\;\underset{k^{\prime }}{\arg\max}\sum_{m=1}^{M}{\left|H_{m,k^{\prime }}\left(n\right)\right|}^{2}=k\\ 0,\;\mathrm{otherwise} \end{array}\right.,\forall k,n,$$
and
(16)$$\rho_{k}^{*}=\bar{E}{\left(\xi \frac{P_{t}}{N}\sum_{n=0}^{N-1}{\left|\sum_{m=1}^{M}H_{m,k^{\prime }}\left(n\right)\frac{H_{m,k^{\prime }\left(n\right)}^{*}\left(n\right)}{\sqrt{\sum_{m^{\prime }=1}^{M}{\left|H_{m,k^{\prime }\left(n\right)}^{*}\left(n\right)\right|}^{2}}}\right|}^{2}+\xi \sigma^{2}\right)}^{-1},\forall k,$$
respectively, where the subscript $k^{\prime }\left(n\right)$ indexes the user, to which the *n*th subcarrier is allocated, according to (Equation 15). Note that if any $\rho_{k}^{*}$ is greater than 1, the minimum required harvesting energy constraint is not met, and a communication outage occurs with that channel impulse response realization, and the obtained secrecy rate per user becomes zero.

Figure 3 shows the cumulative distribution function (CDF) of the lowest total secrecy rate of the proposed algorithm and the minimum effort approach when the transmit power and the minimum required harvesting energy are fixed at $P_{t}=1$ W and $\bar{E}=10$ mW, respectively. The number of realizations is 1000. Both the proposed algorithm and the minimum effort approach do not have an outage for all 1000 realizations. At a probability of 0.5, it can be observed that the secrecy rate of the proposed algorithm is 4.55 bps/Hz, while the secrecy rate of the minimum effort approach is only 2.07 bps/Hz.

As an example of realization, Table 2 shows the channel impulse responses between each transmit (Tx) antenna and each user. User 1 obtains subcarrier 2. User 2 obtains subcarriers 3 and 5. User 3 obtains subcarriers 0, 1 and 4. The power splitting ratios of users 1, 2 and 3 are 0.5425, 0.4051 and 0.6424, respectively. With this optimal solution, the total secrecy rate of users 1, 2 and 3 are 5.5178, 5.5178 and 5.5179 bps/Hz, respectively, and so the lowest total secrecy rate is 5.5178 bps/Hz. This indicates that the optimal solution tends to balance all users to maximize the lowest total secrecy rate. Even though only one subcarrier is allocated to user 1 while three subcarriers are allocated to user 3, the power splitting ratio of user 1 is lower than that of user 3, which means that user 1 has a larger portion of the received signal for information decoding. Moreover, both user 2 and user 3 receive weak signals on subcarrier 2. This helps user 1 achieve a large secrecy rate on subcarrier 2 according to (Equation 7). To observe all cases, Table 3 shows the instantaneous rate of every user ($k^{\prime }$) on every subcarrier (*n*) with beamforming targeting on each user (*k*) according to (Equation 5) and (Equation 6). Figure 4 shows the frequency of subcarrier allocation to each user for 1000 realizations. Each subcarrier is allocated to every user with a similar frequency because the channels between the base station and all users are statistically identical.

The effects of parameter variations on the average of the lowest total secrecy rate are observed in Figure 5, Figure 6 and Figure 7. Figure 5 shows the average of the lowest total secrecy rate as a function of transmit power ($P_{t}$) with the varied minimum required harvesting energy ($\bar{E}$): 10, 30, 50 and 100 mW. It was found that all the curves have the same ceil because the higher transmit power boosts not only the instantaneous rate of the legitimate user but also the instantaneous rate of other users (eavesdroppers). Thus, the secrecy rate, which is the gap of instantaneous rate, is not wider. The proposed algorithm’s ceil is about 3 times higher than the minimum effort approach’s ceil. Another point is that decreasing the minimum required harvesting energy causes the network to reach the ceil at lower transmit power. The reason is that a smaller portion of received power is necessary for energy harvesting, and a larger portion is used for information decoding, in which the same energy is obtained with less transmit power.

Figure 6 presents the average of the lowest total secrecy rate as a function of the minimum required harvesting energy ($\bar{E}$) with the varied transmit power ($P_{t}$): 0.5, 1 and 1.5 W. Every curve has three regimes. Considering a transmit power of 1.5 W and a minimum required harvesting energy of 87 mW, decreasing the required minimum harvesting energy further does not boost the obtained secrecy rate because the network is limited by the transmit power in this regime. That is because every receiver needs the same amount of energy and has a larger portion of received power for information decoding. Without a different surplus, the instantaneous rate gaps among users do not change. When the minimum required harvesting energy is between 87 mW and 320 mW, the obtained secrecy rate decreases as a function of the minimum required harvesting energy. When the minimum required harvesting energy exceeds 320 mW, the network fails to meet the minimum required harvesting energy and obtains zero secrecy rate in this regime. With a transmit power of 0.5 W or 1 W, the curve also has three regimes, a similar trend and the same ceil. The difference is the regimes’ boundaries, which are shifted to the left with lower transmit power. This means that higher transmit power helps the network reach the ceil when the receiver needs high energy to operate. The minimum effort approach gives the similar trend, and reaches the ceil at the same minimum required harvesting energy as that of the proposed algorithm.

Figure 7 presents the average of the lowest total secrecy rate as a function of energy harvesting efficiency (ξ) with the varied minimum required harvesting energy ($\bar{E}$): 10, 30, 50 and 100 mW. It was found that all the curves have the same ceil. This means that further improving the energy harvesting efficiency does not boost the obtained secrecy rate. The lower minimum required harvesting energy helps the network achieve the ceil with lower energy harvesting efficiency. Therefore, the ceil can be reached by either improving the energy harvesting efficiency or using a low power receiver. Also, the minimum effort approach requires higher energy harvesting efficiency than that of the proposed algorithm to reach the ceil.

In a realistic scenario, the network might consist of heterogeneous IoT nodes. The noise variances among nodes are unequal due to hardware differences. The nodes with much higher noise variances might not be able to obtain a non-zero secrecy rate, and the proposed algorithm will fail to find a feasible solution. The aim for future work is to relax the objective by setting a threshold to dismiss a set of IoT nodes so that a feasible solution exists.

## 5. Conclusions

The subcarrier allocation and power splitting ratio are jointly optimized to maximize the lowest total secrecy rate for multi-user SWIPT MISO-OFDMA networks. Since the optimization problem is non-convex, an algorithm has been proposed to find the optimal solution. Compared to the minimum-effort approach, the performance gain is significant. The network performance has a ceil, which can be reached by using our results to design the appropriate transmit power, energy harvesting efficiency and minimum required harvesting energy. In other words, it is not necessary to further increase transmit power and energy harvesting efficiency or to make a receiver use lower power. Moreover, if any factor is a design limitation, other factors can compensate for the limitation to reach optimal performance. 

## Figures and Tables

**Figure 1 sensors-22-03814-f001:**
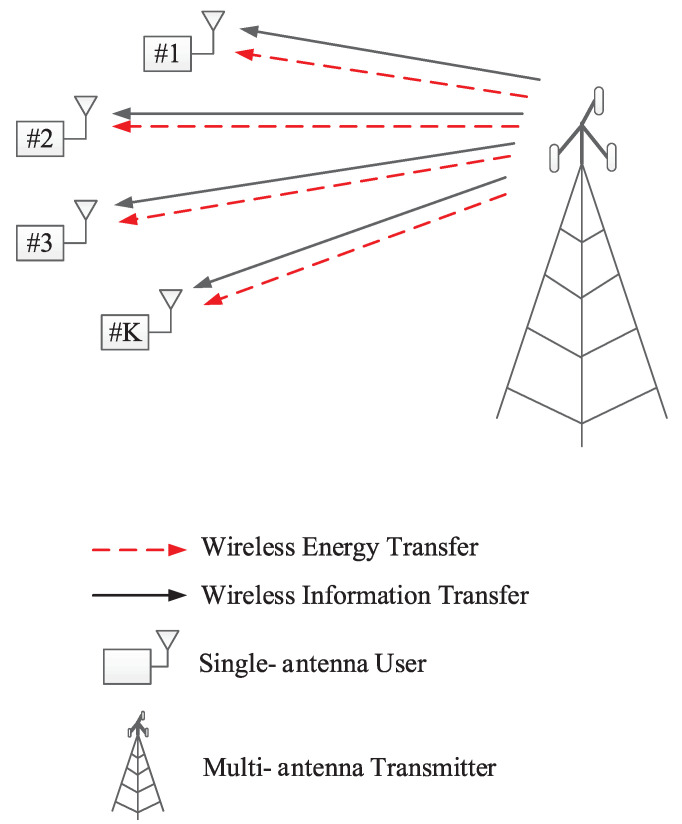
A network model of multi-user downlink communications based on SWIPT.

**Figure 2 sensors-22-03814-f002:**
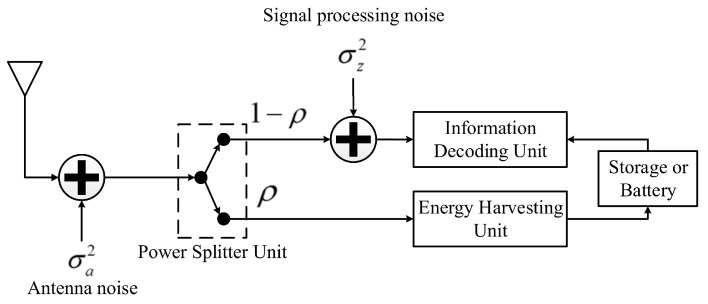
SWIPT receiver model with power splitting scheme.

**Figure 3 sensors-22-03814-f003:**
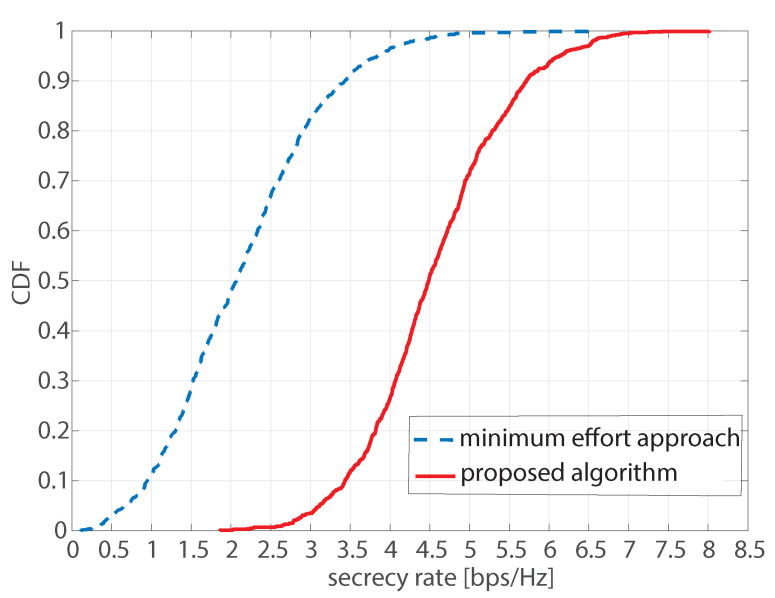
CDF of the lowest total secrecy rate of the proposed algorithm and the minimum effort approach at $P_{t}=1$ W and $\bar{E}=10$ mW.

**Figure 4 sensors-22-03814-f004:**
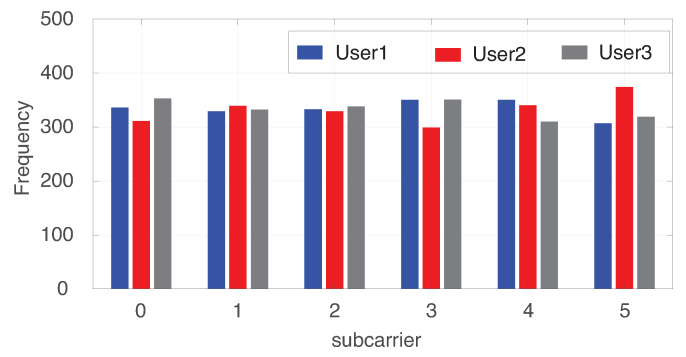
Frequency of subcarrier allocation to each user for 1000 realizations.

**Figure 5 sensors-22-03814-f005:**
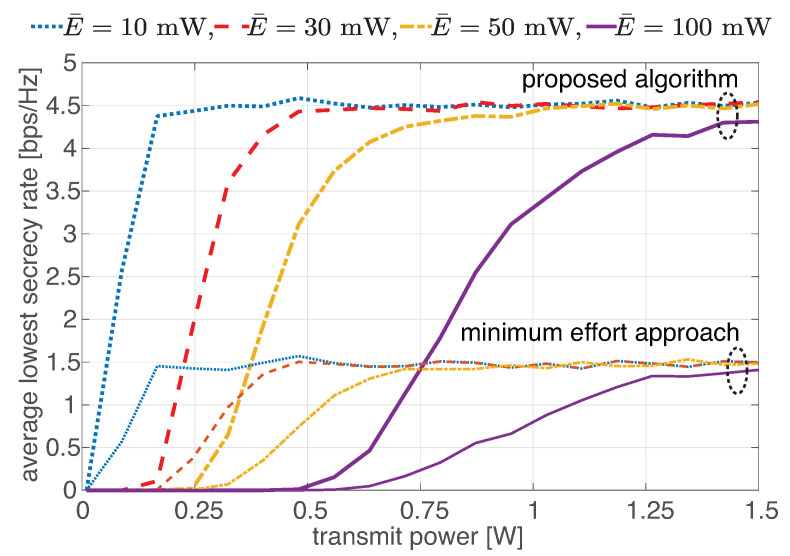
Average of the lowest total secrecy rate as a function of transmit power with varied minimum required harvesting energy.

**Figure 6 sensors-22-03814-f006:**
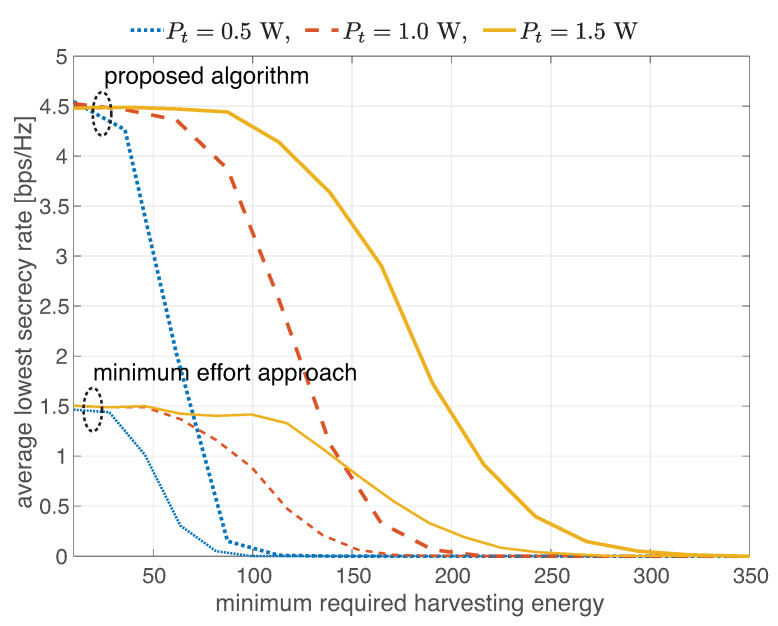
Average of the lowest total secrecy rate as a function of minimum required harvesting energy with varied transmit power.

**Figure 7 sensors-22-03814-f007:**
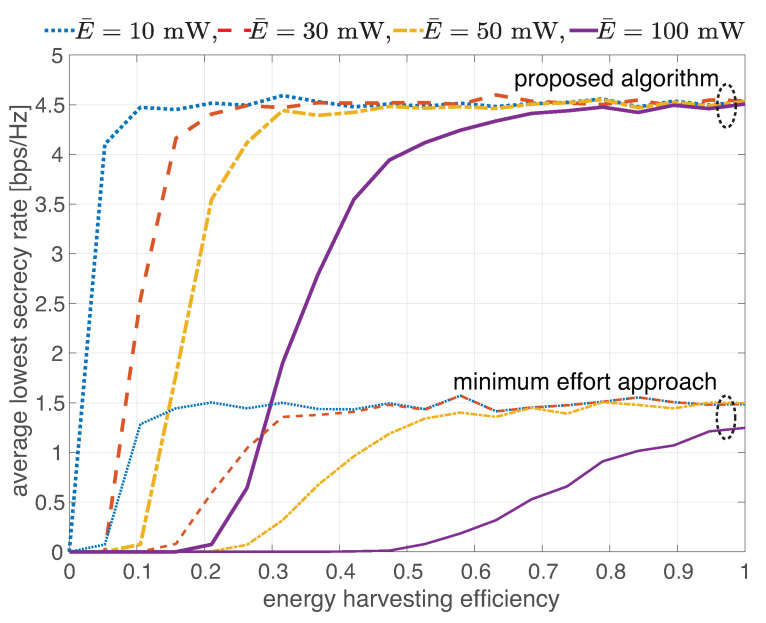
Average of the lowest total secrecy rate as a function of energy harvesting efficiency with varied minimum required harvesting energy.

**Table 1 sensors-22-03814-t001:** Computer simulation parameters.

Parameters	Values
Number of receivers (K)	3
Number of transmitter antennas (M)	4
Number of subcarriers (N)	6
Number of channel delay taps (L)	4
Transmit power $\left(P_{t}\right)$	30 dBm
Noise variance $\left(\sigma^{2}\right)$	−60 dBm
Energy harvesting efficiency (ξ)	0.4
Minimum required harvesting energy $\left(\bar{E}\right)$	10 mW

**Table 2 sensors-22-03814-t002:** A realization of channel impulse responses of every user and transmit antenna.

(User, Tx Antenna)	Channel Impulse Response
(1, 1)	[0.1901 + 0.3865i, 0.3048 + 0.0274i, −0.1533 − 0.0024i, 0.9791 + 0.1313i]
(1, 2)	[0.2565 − 0.3850i, −0.0725 + 0.3891i, 0.4982 − 0.5274i, −0.4269 + 0.8310i]
(1, 3)	[0.1729 − 0.0680i, −0.1073 − 0.4958i, 0.3141 − 0.0627i, −0.2862 + 0.1031i]
(1, 4)	[0.1150 − 0.2844i, −0.6051 − 0.0862i, 0.1129 − 0.4059i, −0.0106 + 0.9141i]
(2, 1)	[0.6484 + 0.3922i, 0.1127 − 0.4293i, 0.1211 + 0.5419i, −0.4773 − 0.0798i]
(2, 2)	[−0.0223 + 0.0115i, −0.0439 + 0.5460i, 0.5011 − 0.2624i, 0.2536 − 0.2176i]
(2, 3)	[0.3658 + 0.3142i, 0.1039 − 0.5029i, −0.4056 − 0.0693i, −1.0410 + 0.0699i]
(2, 4)	[−0.2669 + 0.2463i, −0.0361 + 0.0763i, 0.1106 + 0.0371i, −0.0583 − 0.2358i]
(3, 1)	[−0.7986 − 0.3053i, −0.4623 − 0.3937i, 1.2652 − 0.2721i, 1.0730 + 0.3950i]
(3, 2)	[0.2527 + 0.1953i, 0.5267 + 0.0304i, 0.2374 − 0.3753i, 0.5764 + 0.2645i]
(3, 3)	[0.2570 − 0.2704i, −0.2783 + 0.1726i, −0.3779 + 0.5018i, 0.5085 + 0.5613i]
(3, 4)	[0.4845 + 0.2952i, −0.0854 − 0.4122i, −0.3058 + 0.2554i, 0.2219 + 0.0662i]

**Table 3 sensors-22-03814-t003:** A realization of instantaneous rate of every user on every subcarrier with beamforming targeting on each user.

(Target User, Subcarrier)	User 1	User 2	User 3
(1, 0)	28.5147	26.4973	27.3826
(1, 1)	28.6151	27.3866	27.3654
(1, 2)	29.6687	24.1285	24.1508
(1, 3)	30.6426	25.5290	24.6136
(1, 4)	28.8893	27.6447	27.6087
(1, 5)	28.9884	24.8319	26.3055
(2, 0)	26.6048	28.4072	27.8209
(2, 1)	26.7712	29.2305	29.3225
(2, 2)	25.1895	28.6076	26.8440
(2, 3)	26.2946	29.8769	25.8516
(2, 4)	27.2536	29.2804	28.6473
(2, 5)	23.3430	30.4773	28.5417
(3, 0)	26.7859	27.1168	29.1113
(3, 1)	25.4166	27.9891	30.5640
(3, 2)	24.6454	26.2776	29.1740
(3, 3)	27.4069	27.8792	27.8493
(3, 4)	27.0599	28.4897	29.4381
(3, 5)	25.3984	29.1234	29.8956

## Data Availability

Data are contained within this article.

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
