# Peer review of "Simultaneous Wireless Information and Power Transfer in Multi-User OFDMA Networks with Physical Secrecy"

_sensors, 2022, doi:10.3390/s22103814_

Round 1
Reviewer 1 Report
This paper aims at the communication security of SWIPT in OFDMA network,proposes an algorithm to maximize the secrecy rate of IoT nodes by jointly optimizing the power splitting ratio and subcarrier allocation. The simulation results show that the obtained secrecy rate depends on the base station’s transmit power, the minimum harvested energy requirement of an IoT node and the energy harvesting efficiency.
Modification comments:
- The Abstract part needs to highlight the theme, reduce too many irrelevant introductions, and increase quantitative performance comparison introductions;
- Compress the introduction part, and the wording needs to be refined to reduce the lengthy introduction of technology;
- Although the algorithm performance of the minimum effort approach has been compared in line 190, it is suggested to add the performance comparison of the minimum effort approach in the use case below;
- It is suggested to adjust the position of Figure 1 and Figure 2 to line 124;
- It is suggested to adjust the writing method and font size of the fraction, such as line 120.
Author Response
1. The Abstract part needs to highlight the theme, reduce too many irrelevant introductions, and increase quantitative performance comparison introductions;
We would like to thank the reviewer for guiding us to improve the abstract. The revised abstract focuses on the considered model, the algorithms and the results, as shown in Lines 1-22.
2. Compress the introduction part, and the wording needs to be refined to reduce the lengthy introduction of technology;
Compression and wording refinement were conducted in the revised introduction part in Lines 27-92. Also, the references are correspondingly added and removed.
3. Although the algorithm performance of the minimum effort approach has been compared in line 190, it is suggested to add the performance comparison of the minimum effort approach in the use case below;
The minimum effort approach is added in Figs. 5 (page 11), 6 and 7 (page 12). Also, discussions are deleted/added in lines 240-243, 266-267, 287-289, and 296-298.
4. It is suggested to adjust the position of Figure 1 and Figure 2 to line 124;
Figure 1 and Figure 2 positions were adjusted to line 150, which is next to the paragraph that refers to them.
5. It is suggested to adjust the writing method and font size of the fraction, such as line 120.
The fraction writing was adjusted in lines 143 and 166 (the exponent in equation 3).
Reviewer 2 Report
This manuscript demonstrates the simulation results of a proposed algorithm for optimizing the physical secrecy rate of simultaneous wireless information and power transfer (SWIPT) in orthogonal frequency-division multiple access (OFDMA) networks, illustrating the parameter study findings of the proposed algorithm. The manuscript is well structured. The contents are rich in establishing algorithm and verification work. But some points as indicated below have to be addressed for improving the manuscript:
- The parameters and their values chosen for simulation in Table 1, particularly the Pt, σ2, ξ, and E-hyphen-on-top, must be explained and rationalized.
- This study assumes a constant noise variance for all subcarriers (Table 1), which simplifies the transmit power and CSIs (as indicated in Page 3 Lines 118-120 & 128-130). However, in a realistic environment the overall noise is complex and multiple. Moreover, the parameters of Pt , σ2 and ξ are correlated with one another. The realistic role of noise, which is the key to physical secrecy, is not clearly stated for the simulation. The authors have to duly address that.
- What if the transmitter cannot maximize the instantaneous SNR when the noise becomes a major variable for all individual subcarriers due to hardware differences? The authors have to discuss such a realistic scenario prior to making the conclusions.
- Abbreviation not been defined: P2 Line 39 MIMO, and P3 Line 116 MISO.
- Typo in P5: instantneous.
Author Response
1. The parameters and their values chosen for simulation in Table 1, particularly the Pt, σ2, ξ, and E-hyphen-on-top, must be explained and rationalized.
The explanation and rationale of the chosen simulation parameters are given with references below. We added the explanation and the references in the revised manuscript in P8 Lines 210-217.
- SWIPT in the literature assumes Pt = 1 W (30 dBm) [R1]. Commercial base stations and access points with Pt = 1W are available. Therefore, Pt is set at 30 dBm. ([R1] L. Liu, R. Zhang and K. Chua, "Secrecy Wireless Information and Power Transfer With MISO Beamforming," in IEEE Transactions on Signal Processing, vol. 62, no. 7, pp. 1850-1863, April 1, 2014, doi: 10.1109/TSP.2014.2303422.)
- The power sensitivity for information receiving is - 60 dBm [R1]. Therefore, the noise variance σ2 is set at -60 dBm. ([R1] L. Liu, R. Zhang and K. Chua, "Secrecy Wireless Information and Power Transfer With MISO Beamforming," in IEEE Transactions on Signal Processing, vol. 62, no. 7, pp. 1850-1863, April 1, 2014, doi: 10.1109/TSP.2014.2303422.)
- The experimental self-resonant coils can wirelessly transfer power over distances longer than 2 meters with 40% efficiency [R2]. Therefore, the energy harvesting efficiency ξ is set at 0.4. ([R2]T. H. Abdelhamid, A. Elzawawi and M. A. Elreazek, "Wireless Power Transfer Analysis and Power Efficiency Enhancement via Adaptive Impedance Matching Network," 2021 IEEE International Conference in Power Engineering Application (ICPEA), 2021, pp. 91-96, doi: 10.1109/ICPEA51500.2021.9417839.)
- The smart IoT node consumes a power of 11.84 mW during processing (pattern recognition) in real tests [R3]. Therefore, the minimum required harvesting energy (E-hyphen-on-top) is set at 10 mW. ([R3] V. Kartsch, M. Guermandi, S. Benatti, F. Montagna and L. Benini, "An Energy-Efficient IoT node for HMI applications based on an ultra-low power Multicore Processor," 2019 IEEE Sensors Applications Symposium (SAS), 2019, pp. 1-6, doi: 10.1109/SAS.2019.8705984.)
2. This study assumes a constant noise variance for all subcarriers (Table 1), which simplifies the transmit power and CSIs (as indicated in Page 3 Lines 118-120 & 128-130). However, in a realistic environment the overall noise is complex and multiple. Moreover, the parameters of Pt , σ2 and ξ are correlated with one another. The realistic role of noise, which is the key to physical secrecy, is not clearly stated for the simulation. The authors have to duly address that.
The scope of this paper optimizes subcarrier allocation and power splitting ratio with two motivations, which associate with OFDMA and SWIPT. First, the channel strengths of users on each subcarrier are random. Different subcarrier allocations result in different total secrecy rate of each user. This is an opportunity to assist the user with the lowest total secrecy rate. Second, channels are fixed by the environment, which cannot be changed by the system to assist the user with poor channels, but the power splitting ratios of the stronger users can be increased (lowering their rate) so that the poor user obtains higher total secrecy rate. Optimizing the power allocation certainly improves the total secrecy rate, but it is non-convex and is simplified to solve [R4]. This paper aims to provide a non-simplified solution for benchmarking. Accordingly, the power allocation problem is not included, and the transmit power is divided equally for all subcarriers. Similarly, CSIs can change all the time due to nodes’ mobility, but the problem will be complex and need a simplification. Therefore, we assume that the nodes’ mobility is relatively slow compared with a frame rate of OFDMA, and CSIs become constant for a period of OFDMA frame. We added the explanation and reference in P4 Lines 143-145 and P4-P5 Lines 153-156. ([R4] J. Tang et al., "Optimization for Maximizing Sum Secrecy Rate in SWIPT-Enabled NOMA Systems," in IEEE Access, vol. 6, pp. 43440-43449, 2018, doi: 10.1109/ACCESS.2018.2859935.)
Unequal noise variances indeed affect the total secrecy rate of each user and can be taken into account in equations (4)-(6) without modifying the proposed algorithm. There will be more simulation parameters (different noise variances), but this paper aims to show that the ceil can be reached with three parameters: transmit power, energy harvesting efficiency and minimum required harvesting energy, which can be changed by the system. Accordingly, equal noise variance is set for all subcarriers to exclude the effect of unequal noise variances. We addressed the expected results of unequal noise variances and the transparency of the proposed algorithm in P8 Lines 218-225.
3. What if the transmitter cannot maximize the instantaneous SNR when the noise becomes a major variable for all individual subcarriers due to hardware differences? The authors have to discuss such a realistic scenario prior to making the conclusions.
Heterogeneous IoT nodes certainly have different noise variances and will give different results and new insight. With the large difference between nodes, the very poor nodes will have difficulty achieving a non-zero secrecy rate, and the objective to maximize the lowest secrecy rate among users might be infeasible. In that case, the objective must be relaxed to dismiss very poor users under a threshold, which can be future work. We discuss this point in P11-P13 Lines 299-304.
4. Abbreviation not been defined: P2 Line 39 MIMO, and P3 Line 116 MISO.
MIMO and MISO were defined at P2 Line 55 and P3 Line 139, respectively.
5. Typo in P5: instantneous.
The typo was corrected in P5.